# Left and Right Cortical Activity Arising from Preferred Walking Speed in Older Adults

**DOI:** 10.3390/s23083986

**Published:** 2023-04-14

**Authors:** Julia Greenfield, Véronique Delcroix, Wafae Ettaki, Romain Derollepot, Laurence Paire-Ficout, Maud Ranchet

**Affiliations:** 1Laboratory of Industrial and Human Automation Control, Mechanical Engineering and Computer Science, UMR 8201—LAMIH, University Polytechnic Hauts-de-France, F-59313 Valenciennes, France; 2Health, Safety and Transport Department, Laboratory Ergonomics and Cognitive Sciences Applied to Transport (TS2-LESCOT), University Gustave Eiffel, The French Institute of Science and Technology for Transport, Development and Networks (IFSTTAR), University of Lyon, F-69675 Lyon, France

**Keywords:** fNIRS, walking speed, k-means clustering, gait control, prefrontal activity, cortical activation

## Abstract

Cortical activity and walking speed are known to decline with age and can lead to an increased risk of falls in the elderly. Despite age being a known contributor to this decline, individuals age at different rates. This study aimed to analyse left and right cortical activity changes in elderly adults regarding their walking speed. Cortical activation and gait data were obtained from 50 healthy older individuals. Participants were then grouped into a cluster based on their preferred walking speed (slow or fast). Analyses on the differences of cortical activation and gait parameters between groups were carried out. Within-subject analyses on left and right–hemispheric activation were also performed. Results showed that individuals with a slower preferred walking speed required a higher increase in cortical activity. Individuals in the fast cluster presented greater changes in cortical activation in the right hemisphere. This work demonstrates that categorizing older adults by age is not necessarily the most relevant method, and that cortical activity can be a good indicator of performance with respect to walking speed (linked to fall risk and frailty in the elderly). Future work may wish to explore how physical activity training influences cortical activation over time in the elderly.

## 1. Introduction

Walking is an everyday activity and is considered as one of the bases of human locomotion. It is acquired at a young age of around 11 to 14 months and is refined until the age of around 7 or 8 years [1]. This activity requires the coordination of multiple muscle groups across the lower limbs and lower back to control parameters such as walking phase, step length, step width, and pelvic inclination [2,3,4]. Furthermore, the variability of gait parameters such as double support time, step length [5], and step time [6] has been linked to fall risk in the elderly. The act of walking also provokes cerebral activation: the prefrontal cortex (PFC), in particular the dorso-lateral PFC (DLPFC), is highly active in the control of voluntary movement and is responsible for movement execution [7,8]. Older adults may show an increased PFC activity in complex walking conditions such as dual-task conditions compared to younger adults [9,10].

The effect of aging on walking is a natural phenomenon and alters gait pattern through kinematic changes such as reduced step length and width, decreased walking speed, and lower cadence [6,11,12,13]. These alterations in walking parameters were interpreted as methods to preserve balance during walking by lowering the activity’s complexity and efficiency.

Biomechanical adjustments made to the gait pattern with aging can be considered as tell-tale signs of fall risk. In particular, walking speed is often used as an indicator of fall risk [5,14,15,16,17,18], as well as stride length, step frequency, and double support time [5,14,17]. In addition to walking parameters, emerging evidence has shown that brain activity during walking can predict falls in aging [19]. Brain activation can reveal subtle changes in brain function that may precede behavioural modifications. Cerebral activity during walking can be measured using techniques such as portable electroencephalography (EEG) or functional near-infrared spectroscopy (fNIRS) [10,20,21,22]. Furthermore, a recent study showed that dual-task walking compared to simple-task walking can increase DLPFC activity up to two-fold [10]. 

While walking performance is seen to decrease (with respect to its speed and control) and the required cognitive activity for carrying out a task is seen to increase with advanced age, the trend is not always necessarily linear. Studies comparing gait variables between different age groups within an elderly population do not always conclude high or significant differences [10,23,24,25]. Some studies showed that individuals age at different rates, and that each person has an “ageotype”, that is: a combination of molecular and physiological factors that influence how the body ages [26,27]. The latter article warrants the notion of categorising individuals by factors other than their age based on birth date. Furthermore, a review carried out by Yogev-Seligmann et al. [8] highlighted the fact that there is a high variability in frontal lobe alterations and decline in executive function depending on age, but also education and individual lifestyle; all of which indicate that cognitive capacity degrades with age; however, there is a high between-individual variability. 

Although PFC activity during preferred-speed walking was well investigated [28,29], few studies explored changes in cerebral activity during fast walking compared to slow walking. There is, however, little consensus on whether walking speed affects PFC activity; age may be an influencing factor. A recent study by de Belli et al. [30] found that both age and walking speed affect pre-frontal cortex activity. Their study measured PFC activity in young (mean age = 22 years) and older (mean age = 67 years) adults walking at preferred and fast-walking speeds. Results showed that walking speed did not alter PFC activity in young adults, but this was not the case in older adults who showed a significantly higher increase in PFC activity in the left hemisphere (LH) during fast walking compared to walking at a preferred speed. Harada et al. [31] and Metzger et al. [32] found that fast-speed walking induced greater PFC activity changes in the LH in both young and old adults. Work from two other studies [33,34] found, however, no significant changes in young adults dependent on walking speed. The abovementioned studies indicate that the LH becomes more active in fast walking conditions; hence, this emphasises the notion that the LH and RH of the brain function independently from one another, in particular in more complex situations. Furthermore, a complex situation for an older adult is not necessarily complex for a younger adult.

The aim of this study is, therefore, to analyse left and right cortical activity changes in elderly adults regarding their walking speed. This latter variable will be assessed independently of participants’ age since, as aforementioned studies showed, their relationship is not necessarily linear.

## 2. Materials and Methods

Data from 50 healthy older adults were used in this study. Data were obtained from prior studies [10,20] for which ethical approval was previously obtained by the French biomedical ethics committee (Comité de Protection des Personnes Nord Ouest III Ref. CPP: 2018-01 N° ID RCB: 2017-A03187-46). Informed consent was obtained from all participants prior to the study. Inclusion criteria were: to be able to walk for at least 20 min, and absence of other medical conditions that may interfere in gait. A brief description of materials and methods of the walking task are described below, further information can be found in Hoang et al. [10] and Ranchet et al. [20].

Of the 50 participants whose data were used in this study, the mean age ± standard deviation (SD) was 67 ± 6.8 years old (range: 54–87); data were collected from 31 women and 19 men.

Participants were equipped with a 16-optode fNIRS system (NIRSport, NIRx Medical Technologies, New York, NY, USA), consisting of eight sources and six detectors separated by ~30 mm, placed over the DLPFC (see Figure 1a). Channels 9 and 10 were not considered as these were short separation channels used to quantify peripheral signal artefacts. This device measured the relative concentration of oxygenated (HbO) and deoxygenated (HbR) blood flow to the DLPFC. Relative changes in HbO and HbR concentration were based on changes to DLPFC blood flow compared with measures taken during quiet standing prior to activity onset.

Biomechanical gait parameters were also collected using two wearable sensors, one attached to each shoe (Physiolog^®^5, Gait Up, Lausanne, Switzerland). This device was previously used in studies [10,20,35] for gait analysis. Walking performance measures included gait cycle time (seconds), cadence (steps per minute), stride length (m), and walking speed (m/s). The parameters are expressed through their mean and coefficient of variation (CV). The CV stated is the mean CV of all five walking trials for each participant. 

After being equipped with the fNIRS and Gait Up devices, the protocol was explained to the participant and consisted of the following: the procedure began with 45 s of quiet standing, followed by a walking trial lasting 30 s. A total of five walking trials were carried out. The duration of the rest period between trials varied from 25 to 35 s to diminish possible resonance effects [36]. During these five trials, the participants walked at a comfortable speed around an oval-shaped track marked onto the floor (see Figure 1b). All trials took place indoors, in a well-lit room with minimal visual and auditory disturbances. The walking was performed in the same direction for each of the 5 trials, and the results obtained were averaged.

The following sections present the initial variables and data pre-processing, and finally, the resulting list of variables used for analysis.

### 2.1. Initial Variables and Pre-Processing

Figure 2 shows (a) a typical fNIRS haemodynamic response function where zero represents the mean level of haemoglobin concentration during the quiet standing prior to activity. The data used to generate this graph do not stem from the present study, but from Scholkmann et al. [37]. An increase in oxyhaemoglobin concentration (HbO) occurred in conjunction with a smaller decrease in deoxyhaemoglobin concentration (HbR), as oxygenated blood flowed into the active brain region. The sum of HbO and HbR is the total haemoglobin concentration (HbT); (b) the variable avg_HbO (resp. HbR) represents the average of HbO (resp. HbR) during the first 20 s of the walking trial, while max_HbO (resp. min_HbR) represents the maximum value of HbO (resp. minimum value of HbR) during the same period. Two additional variables combine HbO and HbR: the variable deltaH represents the difference between the highest value of HbO and the lowest value of HbR, and diff_Hb represents the difference between the mean value of HbO and the mean value of HbR. Both HbO and HbR can have positive and negative values; as a consequence, the variables diff_Hb and deltaH can also have negative values. 

The variables used in our study relate to several characteristics of HbO and HbR flow over 20 s from the onset of walking. Change in oxy/deoxy-haemoglobin (HbO and HbR, respectively) refers to the change in haemoglobin levels from the period of quiet standing carried out before the task, for which the reference values were HbO = 0 and HbR = 0. The cerebral activity indicators were averaged over the 5 trials, given for each channel. When data presented >3 extreme values within a channel, data from the given channel were removed. Overall, data from 34 channels were omitted out of 800 (16 channels × 50 persons). The variability of HbO and HbR for a given person can be very high depending on the different fNIRS channels. This variability is visible on Appendix A.

Table 1 defines the selected characteristics for HbO and HbR. All those features were obtained over the time interval [0, 20] seconds and for each of the 16 channels. All cortical activity variables, other than the times to maximal change, were measured in micromole per litre (µmol/L). Time to maximal change variables are given in seconds (s).

### 2.2. Additional Variables

In addition, as is visible in Figure 2, the cerebral activity corresponded to the conjunction of an increase in oxyhaemoglobin concentration with a smaller decrease in deoxyhaemoglobin concentration. Some combinations were proposed in the literature, in addition with HbT [35,36]. To have a better view of how to combine HbO and HbR to analyse cerebral activity, we defined additional variables that combine HbO and HbR (see Table 2).

To acquire comprehensible results from the 18 channels available on the fNIRS device, we analysed which channels to consider and how to regroup them. Of the 18 channels available, only data from channels 1–8 and 11–18 were considered. Channels 9 and 10 were short separation channels.

As aforementioned, previous research highlighted an independence between the left and right cerebral hemispheres, particularly in older adults [30,31]. Based on these previous works, the decision was made to also separate our data between LH and RH.

Despite previous work justifying separation of the LH and RH, their independence was verified through correlation analyses. The variable used for these analyses was the mean relative change in oxyhaemoglobin, as this is the most widely cited and accepted variable for cortical activation [36,38]. Scatter plots between all channels were produced to check for a linear distribution and, based on this, Pearson correlation coefficients using a pairwise method and *p*-values were calculated for each channel. Stronger and more significant correlations were found between the LH channels and between the RH channels, while more moderate correlations were seen between the LH and RH channels. Results from these a priori analyses can be found in Appendix A. 

### 2.3. Complete List of Variables

Collected data were primarily made up of cortical activation indicators and gait parameters. Variables relating to cognition were also obtained from questionnaires carried out prior to the experiment. A comprehensive list of all measures and abbreviations is detailed below for cortical activation (Table 3) and gait parameters (Table 4). Two additional variables were measured for all participants: the fear of falling in the community-dwelling population through the Falls Efficacy Scale—international short version (FES-I_Score; [39]); global cognitive state was assessed by the Montreal Cognitive Assessment (MoCA; [40]).

The coefficient of variation calculated was the ratio of the standard deviation to the mean. In this article, the CV stated is the mean CV of all participants for all gait cycles collected over the five walking trials.

### 2.4. Definition of Two Groups in the Population by K-Means Clustering

Data were sorted based on their walking speed using a K-means clustering algorithm splitting them into one of two groups. Participants were allocated to either slow- or fast-walking speed clusters. Through this partitional clustering method, each participant was assigned to one group only. K-means clustering separates data into a pre-defined number of groups (clusters). Clusters are formed by allocating data points such that, at the end, the sum of the squared error is minimal (squared Euclidean distance between each data point and the cluster centroid) for each cluster. In the case of this study, the elbow method showed that the choice of two clusters provides the best compromise to get less variations within clusters and clusters as different as possible. Appendix A shows the inflection point when using two clusters.

Mean values for all variables were calculated for slow and fast clusters.

The K-means clustering algorithm generated two clusters of participants of n = 22 for the slow cluster and n = 28 for the fast cluster. It can be seen in Table 5 that the distribution of the participants in the slow and fast clusters did not follow the age distribution. Two thirds of the 25 oldest participants, whose ages were above 67, were in the fast cluster, and about two thirds of the 22 participants in the slow cluster were less than 65. Men and women were fairly evenly distributed in the slow and fast clusters. The slow cluster had an average speed of 0.95 m/s, whereas the fast cluster average speed was 1.28 m/s. No participant was considered as severely overweight or obese, which could have changed the values of the walking parameters.

### 2.5. Statistical Methods

A chi² test was used to test for differences between the number of participants and spread of data in each cluster. Data for each cluster were tested for normality using the Shapiro–Wilk test. Following the results from the latter test, either a Student’s independent *t*-test or a Mann–Whitney U test was carried out to test for differences between the slow and fast clusters for a given variable. This was performed for all variables. Further statistical analyses were performed to test for difference between LH and RH cortical activity for a given variable within a cluster. This was carried out using either paired *t*-tests or a Wilcoxon’s signed rank test, for normal or non-normal data, respectively. Effect sizes were calculated as Cohen’s *d* where 0.2 < *d* < 0.5 was considered small, 0.5 ≤ *d* < 0.8 was considered medium, and *d* ≥ 0.8 was large.

All statistical analyses were performed in Matlab (version 2022a, The Mathworks Inc., Natwick, MA, USA); the threshold for statistical significance was set to 0.05.

## 3. Results

### 3.1. Comparisons between Slow and Fast Clusters

The chi-square goodness of fit test revealed no significant difference between the proportion of participants in each cluster, χ² (1,50) = 2.92, *p* = 0.09. Significant differences were observed for some LH cortical activity variables, with the slow cluster showing greater cortical activation than the fast cluster; in particular for variables: avg_HbO_l (+0.092 μmol/L), max_HbT_l (+0.010 μmol/L), diff_Hb_l (+0.082 μmol/L), and tmax_HbO_l (+3.14 s) (see Table 6 and Figure 3).

As expected, cadence, stride length, and walking speed were lower in the slow cluster than in the fast cluster. Walking cadence was 16 steps per minute smaller, stride length was 0.17 m shorter, and walking speed was 0.33 m/s (1.19 km/h) slower. In the slow cluster, gait cycle time was 0.17 s greater. Furthermore, the cadence variability, as measured by cad_CV, was greater in the slow cluster than in the fast cluster by 0.56%. No significant between-group differences were found for the FES-I score and MOCA score. 

### 3.2. Comparisons between Left and Right Cortical Hemispheres

Results from statistical analyses between cortical hemispheres for data within a given cluster showed no significant differences for the slow cluster. However, in the fast cluster, a significant increase in cortical activity in the RH compared to the LH was observed for multiple variables, such as avg_HbO (+0.119 μmol/L), max_HbO (+0.108 μmol/L), diff_Hb (+0.0932 μmol/L), avg_HbT (+0.144 μmol/L), max_HbT (+0.128 μmol/L), deltaH (+0.0944 μmol/L) tmax_HbO (+2.04 s), and tmax_HbT (+1.87 s). A full summary of all cortical activity variables is given in Table 7.

A summary of all statistical results for slow and fast cluster comparisons for cortical activity is given in Figure 3. It presents the values of the main indicators of cerebral activity for left and right hemispheres and slow and fast clusters. Significant differences are signalled with the symbols * and ** for *p*-values under 0.05 and 0.01, respectively.

## 4. Discussion

The aim of this study was to analyse left and right cortical activity changes in elderly adults regarding their walking speed during a simple walking task. Data from 50 participants in the age range of 54–87 years old were analysed and split into two clusters based on their walking speed. It came to our attention that creating groups of data based on age is not always an appropriate method, as advancing in age affects people differently depending on their previous and current lifestyles [26], and on an assessment of certain physical and neural capacities through screening such as the physiological profile assessment [41,42]. To the authors’ knowledge, this is the first study of its kind to use a clustering algorithm to group participants, rather than the more traditional method of separating them by age (for more details on the K-means methodology commonly used, see recent literature reviews [36,43,44]). Furthermore, this study sought to analyse a number of cortical variables which are less commonly cited, such as HbT, deltaH, and diff_Hb; the most frequently calculated variables being related directly to HbO or HbR. 

The data used in this study is a subset of that used in Hoang et al. [45], the latter study showing no significant differences (and only minor absolute differences) between participants aged between 55 and 87 years old. Hoang et al. [45] compared gait parameters (stride time, identical to avg_gct in our study, and preferred walking speed) as well as cortical activity (in their study: ΔHbO and ΔHbR) between three age groups: young adults (aged 18–37), middle-aged adults (aged 55–65), and older adults (aged 67–87); while significant differences were detected between the young versus middle-aged, and young versus older adults groups, very few differences were observed between the middle-aged and older adults groups. The latter two groups are those used in the present study and the lack of differences between those from Hoang et al. [45] may also be put down to proximity of the groups in terms of age (only two years separate the cut-off for the latter groups, whereas 18 years separate the young and middle-aged groups). However, results from [45] showed a tendency for a greater increase in DLPFC activity, particularly in the right hemisphere in older adults during simple walking compared to young and middle-aged adults. 

The K-means clustering algorithm created two clusters based on participants’ walking speed: slow cluster and fast cluster for slow and fast walkers, respectively. The main results from our analyses showed that the task of simple walking triggered greater overall prefrontal cortical activity in participants of the slow cluster. Furthermore, within the participants in the fast cluster, cortical activity was significantly greater in the RH than in the LH.

The division of the participants into two clusters was based on their walking speed, the difference in mean walking speed was 0.33 m/s (1.19 km/h). As observed in previous studies [46], a greater walking speed was also associated with a lower (faster) mean gait cycle time [47], a higher cadence [5,47,48,49], and a longer stride length [5,47,48,49]. Concerning variation of gait parameters, expressed through the coefficient of variation (CV), only cadence (cad_CV) demonstrated a significant difference (0.56%, *p* = 0.04) between the slow and fast clusters. Despite the non-significance of other gait parameter CV between the two clusters, there was a tendency to lower CV in the fast cluster. The participants who, on average, had a greater walking speed also demonstrated a more stable gait pattern. Multiple studies previously showed that an increase in walking speed lead to a decrease in gait variability [5,14,50] and also a decrease in inter-segment coordination variability [50,51]. These variables were linked to gait control [52], and our results suggest that the participants in the slow cluster had lower control and stabilisation of their gait. 

In the elderly population, walking speed was associated with the risk of falls [5,14,15]. One may argue that walking speed lessens with aging and that groups should, therefore, be created based on age. This concept was demonstrated by multiple authors [53,54,55]; however, there are also studies which do not find any significant influence of aging on gait parameters [5,56,57]. Furthermore, if age were to be the most influential factor on gait parameters, then the clusters originating from our K-means clustering algorithm would have been differing in age. However, chi-square results demonstrated no significant differences between clusters based on age groups. The latter idea was implemented by [58] who divided participants based on walking speed (fast versus slow); these authors did, however, find a significant difference in the mean age of the two groups.

One may argue that the slow cluster naturally showed greater variability in gait parameters as they walked at a slower speed, and this notion was shown by previous studies [59,60]). However, the work from Kang and Dingwell [61] suggested that variability in gait parameters in elderly individuals walking at their preferred speed may be due to reductions in internal characteristics such as muscle strength and joint flexibility. The latter parameters can be associated with age-related changes which, again, support the notion that people age at different rates and that the definition of an ageotype for participant categorisation may, at times, be more appropriate.

Cortical activity was seen to increase in participants within the slow cluster compared to the fast cluster for the avg_HbO, avg_HbT, and diff_Hb variables. Time to maximum oxyhaemoglobin was also greater in the slow cluster. This indicates that the cluster of participants who were considered to be “fast” walkers reached their maximum necessary oxyhaemoglobin level faster than the slow cluster and also observed a lower brain oxygenation. Other cortical variables (max_HbO, max_HbT, and deltaH) demonstrated a higher increase in activity, although this difference was not significant. The act of walking at a self-selected comfortable pace is, therefore, seen to be more demanding, on a cortical level, for participants in the slow cluster [62]. This finding is linked to the compensation-related utilization of neural circuits hypothesis (CRUNCH), which is usually associated with increasing age being related to an increase in the required cortical activity for the execution of a given task [63]. However, in our study, CRUNCH would apply to participants within a similar age bracket and implies that it is not only age that influences the use of compensatory neural networks. 

The CRUNCH model highlights two types of compensation based on cortical activity and performance: successful and unsuccessful [63]. Within our study, we can consider that the slow cluster carried out unsuccessful compensation as they required greater cortical activity to achieve a slower walking speed compared to the fast cluster. 

The comparison of cortical activity changes between the LH and RH for participants within a given cluster showed significantly higher increase in cortical activity in the RH within the fast cluster only. Yet, in the case of multiple cortical variables (avg_HbO, avg_HbT, diff_Hb), this greater increase in RH activation in the fast cluster was nonetheless lower or equal to the RH activation change in the slow cluster (see Figure 3 and Table 7). Change in LH activation was also lower in the fast cluster compared to the slow one. These differences support evidence from previous studies [9,45], and highlight the possibility that the increased LH and RH activation in the slow cluster is a compensatory activity.

Previous studies, such as De Belli et al. [30], Metzger et al. [32], and Harada et al. [31], showed that within-subject differences between fast- and preferred-speed walking generates greater changes in cerebral blood flow to the left hemisphere. However, the results from our study showed an increase in blood flow to the right cerebral hemisphere, compared to the left hemisphere, in the cluster who displayed a greater walking speed. Nonetheless, a higher overall cortical activation increase (left and right hemispheres) was observed in participants producing a slower spontaneous walking speed. Participants demonstrating a greater walking speed were not asked to walk at a slower speed, or vice versa. Hence, differences between our study and those mentioned above may be due to differences in the protocol with respect to the required walking speed.

Certain studies [20,64] suggested that loss of motor autonomy in older adults may be detected through simple-task walking. The automaticity of walking in adults decreases with age; however, its rate of decline varies depending on the individual [26,27]. As automaticity processes in the brain are reduced for the performance of a simple walking task, the required cortical activity increases, as does the risk of falling [29,35]. 

In the present study, participants who required greater executive resources (shown through increased DLPFC activity) also demonstrated a gait pattern that tends towards the profile of an individual with a greater fall risk [5,6], namely through slower gait speed, lower cadence, decreased stride length, and increased gait cycle time, as well as increased variability in all of these aforementioned variables. A large UK biobank study [65] associated faster pace walking with improved life expectancy due to the physical health benefits brought about by a brisker walking speed, such as increased cardiorespiratory and vascular function, musculoskeletal capacity, and advantages on mental health state.

The hemispheric asymmetry reduction in older adults (HAROLD) hypothesis states that older adults present greater bilateral activity than younger adults for a given task [66]. The results from the present study suggest that models/hypotheses such as HAROLD and CRUNCH may go further than just being age-related. Within our study, two clusters of similar mean age showed significant differences in cortical activity and gait parameters linked to fall risk. It may be worth extending both the CRUNCH and HAROLD notions to consider performance variables rather than just age, but also fall risk and frailty.

The work presented in this study showed that elderly adults should not always be categorised based on age, but that other factors, such as walking speed, can be used to differentiate groups. Our results displayed differences in cortical activation between people whose preferred walking speed is considered as slow or fast, those with a slower walking speed requiring greater cortical activation than those with a faster walking speed.

Since walking speed is considered as one of the primary predictors of fall risk in the elderly, it is interesting to note that a simple walking task may also lead to an overload of cerebral functioning (seen through a strong increase in cortical activation). This may help to explain why certain individuals are highly prone to falling when an obstacle must be negotiated [67,68] and may also be related to the type of trajectory required. The work from Belluscio et al. [69] showed that curvilinear trajectories require a greater cognitive demand than linear ones in younger adults. This would also likely be the case in older adults as it leads to task complexity [70]. If a person’s executive function is already saturated, then the appearance of an obstacle (be it physical, visual, or auditory) will lead to an increased fall risk as the individual will not have the necessary resources to mentally process the obstacle. Therefore, future work may wish to build on these ideas by investigating cortical activation changes during obstacle negotiation, and the possible effects that a training program to increase walking speed may have on cortical activity; the latter notion being related to the neural efficiency hypothesis [71]. Activities such as Nordic walking are known to increase joint range of motion and activity levels in the elderly, it may be of interest to assess the cerebral effect of such activities. The effect of a training program on cortical activity was already investigated in elite cyclists and showed that increased training leads to lower cortical demand [72,73], further research may seek to assess these effects in an older population.

Certain limitations can be applied to our study and its methods; namely, the combined analysis of linear and curvilinear trajectories. Human gait is not necessarily managed in the same way depending on the trajectory angle [69] and these analyses should have been treated separately. Furthermore, 50 participants is a relatively small sample size. It would be interesting to repeat the experiment with a greater number of participants to confirm or reject the drawn conclusions.

Gait speed can be influenced by leg length, which is related to height. The height of the participants was not collected during this experiment and, therefore, we are unable to comment on whether this would have influenced the measured gait speeds in each cluster.

## 5. Conclusions

The work presented in this study showed that elderly adults should not always be categorized based on age, but that other factors, such as walking speed, can be used to differentiate groups. The use of clustering algorithms, rather than classic age-based separation, was employed to create population groups. Our results displayed differences in cortical activity between people whose preferred walking speed is considered as slow or fast. The results of the present study suggest that older adults with a slower walking speed need more cognitive resources to perform a given walking task, as compared with older adults with a faster walking speed. Furthermore, the increased prefrontal activation in LH and RH in the slow cluster can be interpreted as a possible compensatory activity. These findings could reflect a higher risk of falling in older adults with a slower preferred walking speed being related not only to biomechanical parameters, but also cortical activity. This hypothesis could be further investigated in future research works, as these outcomes have the potential to help explain fall risk in the elderly. Moreover, for application to the elderly, or patients with neurological disorders, follow-up work may wish to examine the effects of physical training programs on cortical activity.

## Figures and Tables

**Figure 1 sensors-23-03986-f001:**
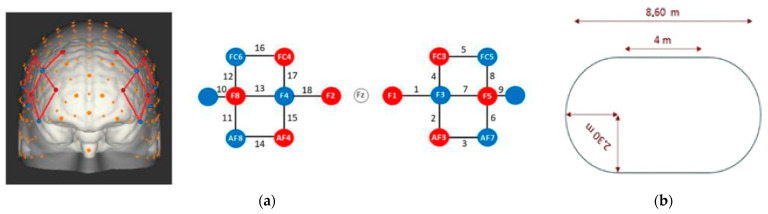
(**a**) fNIRS montage measuring the dorsolateral prefrontal cortex according to the EEG 10-10 system. Red circles represent sources and blue circles represent detectors. Numbers represent channels. (**b**) Walking path configuration. Images adapted from Hoang et al. [10].

**Figure 2 sensors-23-03986-f002:**
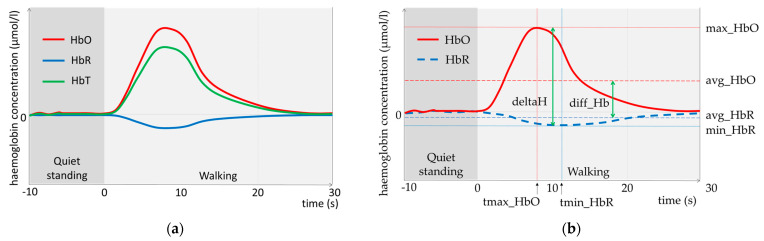
A typical fNIRS haemodynamic response function, adapted from Scholkmann et al. [37], where (**a**) [HbT] corresponds to total haemoglobin concentration (HbO+HbR) and (**b**) max_HbO (resp. minHbR) corresponds to the maximum (resp. minimum) of the oxyhaemoglobin (resp. deoxyhaemoglobin) concentration over the 20 first seconds of walking; avg_HbO and avg_HbR correspond to the average during the same period; diff_Hb corresponds to the difference between avg_HbO and avg_HbR; deltaH corresponds to the difference of max_HbO and min_HbR and tmax_HbO (resp. tmin_HbR) corresponds to the time when the maximum HbO (resp minimum HbR) is reached.

**Figure 3 sensors-23-03986-f003:**
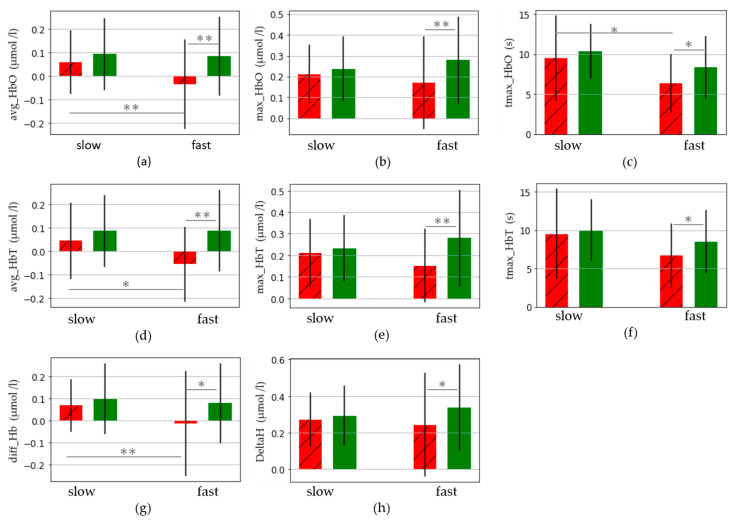
Left (red, striped) and right (green, unstriped) cortical activity in slow and fast clusters for 8 measures: avg, max, and tmax for HbO (**a**–**c**) and HbT (**d**–**f**), diff_Hb (**g**) and delta_H (**h**). Significant differences are signalled with the symbols * and ** for *p*-values under 0.05 and 0.01, respectively.

**Table 1 sensors-23-03986-t001:** Calculated variables for cortical activity based on oxygenated and deoxygenated haemoglobin.

Variable Description	HbO	HbR
the mean value during the first 20 s of the walking trial.	avg_HbO	avg_HbR
the minimum or maximum values during the first 20 s of the walking trial.	max_HbO	min_HbR
time of the minimum or maximum values during the first 20 s of the walking trial.	tmax_HbO	tmin_HbR

**Table 2 sensors-23-03986-t002:** Additional variables combining HbO and HbR, which are relative to the quiet standing phase prior to walking procedure. The HbO and HbR measured during the quiet standing phase are considered to be zero/baseline values.

Abbreviation	Definition	Description
avg_HbT	mean(HbO+HbR)	Average of total haemoglobin during the 20 first seconds of the walking trial.
max_HbT	max(HbT)	Maximum value of HbT during the 20 first seconds of the walking trial.
tmax_HbT	Time at max(HbT)	The time at which max_HbT occurs
diff_Hb	diff_Hb = avg_HbO − avg_HbR	Difference between the average in oxy- and deoxy-haemoglobin as a measure of cortical activation
deltaH	deltaH = max_HbO − min_HbR	Maximum difference in oxy- and deoxy-haemoglobin

**Table 3 sensors-23-03986-t003:** Abbreviation and description of all variables calculated from fNIRS data. The HbO and HbR measured during the quiet standing phase are considered to be zero/baseline values.

Abbreviation	Description of the Cortical Activation Parameters
avg_HbO_l	Mean change in oxyhaemoglobin, left hemisphere
avg_HbO_r	Mean change in oxyhaemoglobin, right hemisphere
avg_HbR_l	Mean change in deoxyhaemoglobin, left hemisphere
avg_HbR_r	Mean change in deoxyhaemoglobin, right hemisphere
avg_HbT_l	Mean change in total haemoglobin, left hemisphere
avg_HbT_r	Mean change in total haemoglobin, right hemisphere
max_HbO_l	Maximum change in oxyhaemoglobin, left hemisphere
max_HbO_r	Maximum change in oxyhaemoglobin, right hemisphere
max_HbR_l	Maximum change in deoxyhaemoglobin, left hemisphere
max_HbR_r	Maximum change in deoxyhaemoglobin, right hemisphere
max_HbT_l	Maximum change in total haemoglobin, left hemisphere
max_HbT_r	Maximum change in total haemoglobin, right hemisphere
diff_Hb_l	Mean difference in oxy and deoxy -haemoglobin as a measure of cortical activation, left hemisphere
diff_Hb_r	Mean difference in oxy and deoxy -haemoglobin as a measure of cortical activation, right hemisphere
deltaH_l	Maximum difference in oxy and deoxy -haemoglobin, left hemisphere
deltaH_r	Maximum difference in oxy and deoxy -haemoglobin, right hemisphere
tmax_HbO_l	Time elapsed at maximal change in oxyhaemoglobin, left hemisphere
tmax_HbO_r	Time elapsed at maximal change in oxyhaemoglobin, right hemisphere
tmax_HbR_l	Time elapsed at maximal change in deoxyhaemoglobin, left hemisphere
tmax_HbR_r	Time elapsed at maximal change in deoxyhaemoglobin, right hemisphere
tmax_HbT_l	Time elapsed at maximal change in total haemoglobin, left hemisphere
tmax_HbT_r	Time elapsed at maximal change in total haemoglobin, right hemisphere

**Table 4 sensors-23-03986-t004:** Abbreviation and description of gait variables.

Abbreviation	Description of the Gait Parameters
avg_gct	Mean gait cycle time, the time from one heel strike to the next heel strike of the ipsilateral foot (stride time)
gct_CV	Coefficient of variation for gait cycle time
avg_cad	Mean walking cadence
cad_CV	Coefficient of variation for walking cadence
avg_Slength	Mean stride length, the distance covered over a given stride
Slength_CV	Coefficient of variation for stride length, the distance covered over a given stride
avg_speed	Mean walking speed
speed_CV	Coefficient of variation for walking speed

**Table 5 sensors-23-03986-t005:** Number of participants by age, gender, and gait speed groups, with mean speed for each cluster given in parentheses.

	55–65 Years Old	67–85 Years Old	Men	Women	Total
slow cluster (0.95 m/s)	14	8	7	15	22
fast cluster (1.28 m/s)	11	17	12	16	28
Total	25	25	19	31	50

**Table 6 sensors-23-03986-t006:** Results of statistical analyses comparing the two clusters based on walking speed. Variables are presented as the mean value for the given cluster ± standard deviation (SD). Effect size is given with lower and upper confidence intervals (CI). Where significant differences lie in the data, the values are shown in bold. Variables related with cortical activities are given in µmol/L, times are given in s. The symbol * (respectively ** and ***) denotes a significant difference with *p* < 0.05 (respectively 0.01 and 0.001). Values in bold indicate that they are significantly higher than their counterpart.

Variable	Slow Cluster (Mean ± SD)	Fast Cluster (Mean ± SD)	*p*-Value	Effect Size (Cohen’s *d*) [Lower CI, Upper CI]
Age (years)	66.18 ± 5.87	67.46 ± 7.62	0.518	−0.18 [−0.75, 0.37]
FES-I_score	20.05 ± 3.75	19.21 ± 3.11	0.340	0.24 [−0.32, 0.8]
MoCA	26.82 ± 2.08	27.14 ± 1.80	0.611	−0.17 [−0.73, 0.39]
**Gait parameters**				
avg_gct (s) ***	**1.24 ± 0.14**	1.07 ± 0.09	**<0.001**	1.42 [0.82, 2.08]
avg_cad (steps/min) ***	96.27 ± 12.11	**112.81 ± 9.41**	**<0.001**	−1.52 [−2.19, −0.91]
avg_Slength (m) ***	1.16 ± 0.08	**1.33 ± 0.11**	**<0.001**	−1.76 [−2.45, −1.12]
avg_speed (m/s) ***	0.95 ± 0.14	**1.28 ± 0.10**	**<0.001**	−2.68 [−3.50, −1.94]
gct_CV (%)	4.42 ± 3.00	3.56 ± 1.53	0.054	0.37 [−0.19, 0.94]
cad_CV (%) *	**4.05 ± 1.65**	3.49 ± 1.45	**0.041**	0.36 [−0.20, 0.93]
Slength_CV (%)	6.65 ± 2.78	6.28 ± 1.46	0.792	0.17 [−0.39, 0.73]
speed_CV (%)	8.08 ± 1.57	7.86 ± 1.81	0.406	0.13 [−0.43, 0.69]
**Cortical activity parameters**				
avg_HbO_l (µmol/L) **	**0.056 ± 0.135**	−0.035 ± 0.190	**0.005**	0.54 [−0.02, 1.11]
avg_HbO_r (µmol/L)	0.092 ± 0.153	0.084 ± 0.168	0.854	0.05 [−0.51, 0.61]
max_HbO_l (µmol/L)	0.212 ± 0.141	0.170 ± 0.221	0.087	0.22 [−0.34, 0.78]
max_HbO_r (µmol/L)	0.238 ± 0.154	0.279 ± 0.211	0.632	−0.21 [−0.78, 0.34]
diff_Hb_l (µmol/L) **	**0.068 ± 0.120**	−0.014 ± 0.238	**0.003**	0.42 [−0.14, 0.99]
diff_Hb_r (µmol/L)	0.098 ± 0.160	0.079 ± 0.181	0.696	0.11 [−0.45, 0.67]
avg_HbR_l (µmol/L)	−0.012 ± 0.047	−0.021 ± 0.071	0.777	0.15 [−0.41, 0.71]
avg_HbR_r (µmol/L)	−0.006 ± 0.036	0.005 ± 0.060	0.462	−0.21 [−0.77, 0.35]
min_HbR_l (µmol/L)	−0.060 ± 0.050	−0.074 ± 0.083	0.792	0.19 [−0.36, 0.76]
min_HbR_r (µmol/L)	−0.055 ± 0.037	−0.060 ± 0.070	0.732	0.07 [−0.48, 0.63]
avg_HbT_l (µmol/L) *	**0.045 ± 0.163**	−0.056 ± 0.161	**0.020**	0.61 [0.05, 1.20]
avg_HbT_r (µmol/L)	0.086 ± 0.154	0.088 ± 0.175	0.961	−0.01 [−0.57, 0.54]
max_HbT_l (µmol/L)	0.211 ± 0.155	0.152 ± 0.170	0.111	0.36 [−0.20, 0.93]
max_HbT_r (µmol/L)	0.232 ± 0.153	0.280 ± 0.224	0.689	−0.24 [−0.80, 0.32]
deltaH_l (µmol/L)	0.272 ± 0.148	0.244 ± 0.283	0.071	0.12 [−0.44, 0.68]
deltaH_r (µmol/L)	0.293 ± 0.162	0.338 ± 0.236	0.689	−0.21 [−0.78, 0.34]
tmax_HbO_l (s) *	**9.47 ± 5.33**	6.32 ± 3.70	**0.018**	0.69 [0.12, 1.28]
tmax_HbO_r (s)	10.38 ± 3.44	8.37 ± 3.90	0.062	0.54 [−0.02, 1.11]
tmax_HbT_l (s)	9.51 ± 5.86	6.66 ± 4.25	0.052	0.56 [0, 1.14]
tmax_HbT_r (s)	10.02 ± 3.96	8.54 ± 4.10	0.113	0.36 [−0.20, 0.93]

**Table 7 sensors-23-03986-t007:** Results of statistical analyses comparing the activity of the left and right cortical hemispheres. Variables are presented as the mean value for the given hemisphere ± standard deviation (SD), the associated *p*-value, effect size based on Cohen’s d, and lower and upper confidence intervals (CI). Where significant differences lie in the data, the values are shown in bold. Variables related with cortical activities are given in µmol/L, times are given in seconds. The symbol * (respectively ** and ***) denotes a significant difference with *p* < 0.05 (respectively 0.01 and 0.001). Values in bold indicate that they are significantly higher than their counterpart.

Variable	LH (Mean ± SD)	RH (Mean ± SD)	LH–RH (Mean ± SD)	*p*-Value	Effect Size (Cohen’s *d*) [Lower CI Upper CI]
	**Slow cluster**				
avg_HbO (µmol/L)	0.056 ± 0.135	0.092 ± 0.153	−0.036 ± 0.147	0.269	−0.24 [−0.69, 0.19]
max_HbO (µmol/L)	0.212 ± 0.141	0.238 ± 0.154	−0.025 ± 0.136	0.391	−0.17 [−0.57, 0.22]
diff_Hb (µmol/L)	0.068 ± 0.120	0.098 ± 0.160	−0.030 ± 0.137	0.315	−0.21 [−0.63, 0.21]
avg_HbR (µmol/L)	−0.012 ± 0.047	−0.006 ± 0.036	−0.006 ± 0.060	0.671	−0.13 [−0.75, 0.48]
min_HbR (µmol/L)	−0.060 ± 0.050	−0.055 ± 0.037	−0.004 ± 0.057	0.935	−0.13 [−0.75, 0.48]
avg_HbT (µmol/L)	0.045 ± 0.163	0.086 ± 0.154	−0.041 ± 0.178	0.291	−0.25 [−0.74, 0.22]
max_HbT (µmol/L)	0.211 ± 0.155	0.232 ± 0.153	−0.020 ± 0.163	0.563	−0.13 [−0.59, 0.32]
deltaH (µmol/L)	0.272 ± 0.148	0.293 ± 0.162	−0.021 ± 0.124	0.438	−0.13 [−0.48, 0.21]
tmax_HbO (s)	9.47 ± 5.33	10.38 ± 3.44	−0.92 ± 5.50	0.443	−0.20 [−0.73, 0.32]
tmax_HbT (s)	9.51 ± 5.86	10.02 ± 3.96	−0.51 ± 5.91	0.689	−0.10 [−0.61, 0.40]
	**Fast cluster**				
avg_HbO (µmol/L) **	−0.035 ± 0.190	**0.084 ± 0.168**	**−0.119 ± 0.148**	**0.001**	−0.65 [−1.01, −0.32]
max_HbO (µmol/L) **	0.170 ± 0.221	**0.279 ± 0.211**	**−0.108 ± 0.147**	**0.002**	−0.49 [−0.78, −0.22]
diff_Hb (µmol/L) **	−0.014 ± 0.238	**0.079 ± 0.181**	**−0.093 ± 0.170**	**0.004**	−0.43 [−0.76, −0.12]
avg_HbR (µmol/L)	−0.021 ± 0.071	0.005 ± 0.060	−0.026 ± 0.070	0.088	−0.38 [−0.80, 0.02]
min_HbR (µmol/L)	−0.074 ± 0.083	−0.060 ± 0.070	−0.014 ± 0.079	0.202	−0.18 [−0.57, 0.21]
avg_HbT (µmol/L) ***	−0.056 ± 0.161	**0.088 ± 0.175**	**−0.144 ± 0.157**	**<0.001**	−0.84 [−1.26, −0.46]
max_HbT (µmol/L) **	0.152 ± 0.170	**0.280 ± 0.224**	**−0.128 ± 0.165**	**0.001**	−0.63 [−0.99, −0.30]
deltaH (µmol/L) **	0.244 ± 0.283	**0.338 ± 0.236**	**−0.094 ± 0.170**	**0.004**	−0.36 [−0.62, −0.10]
tmax_HbO (s) *	6.32 ± 3.70	**8.37 ± 3.90**	**−2.04 ± 4.09**	**0.014**	−0.53 [−0.96, −0.12]
tmax_HbT (s) *	6.66 ± 4.25	**8.54 ± 4.10**	**−1.87 ± 4.60**	**0.040**	−0.44 [−0.8, −0.02]

CI = confidence interval.

## Data Availability

The data presented in this study are available on request from the corresponding author. The data are not publicly available due to continued use of the dataset within this research group.

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
