# Peer review of "Left and Right Cortical Activity Arising from Preferred Walking Speed in Older Adults"

_sensors, 2023, doi:10.3390/s23083986_

Round 1

Reviewer 1 Report

The work proposes an interesting point-of-view and analysis of how older adults' motion (here focused on walking) can be classified by looking at the walking speed (and additional fNIRS, gait, and cognitive data) besides their age.

The structure of the paper follows almost the guidelines for proper communication, with good and fluent English. However, some changes can be made to further improve the readability of the documents. Here are some recommendations:

1. Check if figures and tables are adequately referenced within the text (e.g., Figure 1 is not called anywhere; there are two Table 4 on page 6/18);

2. Check the caption of each figure and/or table to be the most relevant to the represented data (e.g., Table 3. Abbreviation and description of all variables calculated from fNIRS, gait and cognitive data. -> it does not properly describe the table data since both gait and cognitive parameters are not there reported);

3. Tables should be organized to fit in the page space without splitting among pages, which makes their interpretation difficult and misleading;

4. Figures (e.g., Figure 2) can be improved by augmenting their resolution, matching the font (size and type) with the one of the template, and plotting the data by taking into account visual disorders (i.e., using dashed/dotted linea/areas to represent different features instead of using colors);

5. Citations need to follow the same style: a citation can be found both within square brackets and as superscripts in the paper.

6. A Conclusion section is essential at the end of the manuscript to summarize the work context, methodology, and main outcomes.

 Moving now to the technical aspects, four main points could be further discussed:

1. From the introduction and throughout the paper, it is clear what the context is and how the walking analysis has been carried out. However, a final applicative scenario of the paper outcomes is missing: in other terms, how the separation of older adults into two distinct groups (without distinction based on age, walking speed, or additional features) and the related analysis of cerebral activation have an impact on a diagnosis? Or on their quality of life? Or on the clinical evaluation of a therapy? For readers not working in the field, it should be a clear point to state out at the beginning and end of the paper.

2. The authors assess that this work includes a lot of additional parameters w.r.t. other works in the field, which instead focused on just a features subset. The inclusion of additional variables, computed through statics operators (e.g., avg, max, min, diff, delta) on the initial ones, creates a more complex feature space to work on. Anyway, sometimes, the use of derivative variables hides the correlation between them, thus increasing the dataset dimension but without giving any additional information. Did the author check the statistical independency among all the features that have been used or extracted, hence demonstrating their effective role in the analysis?

3. Since the authors affirm that "To the author’s knowledge, this is the first study of its kind to use a clustering algorithm to group participants, rather than the more traditional method of separating them by age.  (page 10/18)", additional details should be provided to confirm the correctness of their K-means model and how it has been validated before proceeding to the further analysis.

4. A comparison table with other similar and inherent works should ease the discussion section, making directly visible the difference in the test setup and outcome between different works and research groups.

Reviewer 2 Report

Functional near-infrared spectroscopy (fNIRS) is a non-invasive alternative neuroimaging technique that allows assessment during actual movement, brain activation while a person performs motor tasks in the environment, as well as during cognitive tasks. The technique alone or in combination with EEG is often used in the functional assessment of healthy individuals as well as those with neurological dysfunctions.

I have carefully read the article entitled "Left and right cortical activity based on preferred walking speed in older adults." I believe that in terms of its subject matter it has great relevance to kinesiology and neurology and rehabilitation. Nevertheless, the article has some imperfections that should be improved.

The research problem is well presented in the introduction section. The purpose of the paper is clearly stated. It follows logically from the well-described background of the study. However, the question of analyzing the dependence of gait speed on the age of the participants is missing, which was thoroughly analyzed in the results and discussed in the discussion.

In the description of the inclusion criteria, I presume that the participants did not have limitations due to orthopedic dysfunctions (past injuries, fractures) that may have affected the gait pattern. Although the authors refer to previous publications in their description of the participants, I believe that the inclusion criteria should be described in detail.

The description of the method of gait assessment using IMU sensors should be described in detail. Also missing is the provision of a reference to previous applications in gait research.

In the paragraph describing the gait assessment procedure, I suggest describing the gait track in detail or adding a link to a previous study in which the same procedure was used. I suggest adding information on whether the gait was done in the same direction in each of the 5 trials, and whether the results obtained were averaged or whether the fastest trial was selected, for example.

It was very interesting how the slow group and the fast group of participants were separated. The number of women and men in each group should be added. Please also include in the description of the participants information about the physique in the slow and fast groups, unless obesity was an exclusion factor from the study.

The discussion is very interesting. The method of dividing the group into slow walkers and fast walkers and the discussion of the relationship between gait speed and age provides new knowledge in gait analysis. However, there is a lack of consideration of other factors that can undoubtedly have a significant impact on gait speed. The discussion, however, is devoid of limitations and conclusions.

In conclusion, the work is in line with the profile and provides new data for gait analysis. The work should be referred for publication after taking into account the above comments.

Round 2

Reviewer 2 Report

Thank you for considering the comments. I believe that the results of the study introduce new knowledge in the field of gait analysis, which is beyond previous facts. I am confident that the results will be useful in further gait analyses, both of healthy subjects and those with neurological disorders.

Author Response

Dear reviewer,

Thank you for your comments and also for taking the precious time to review our study. We sincerely hope that the results will lead to improvements and advances in this area.

Kind regards,

Julia Greenfield